# Lead (Pb) as a Factor Initiating and Potentiating Inflammation in Human THP-1 Macrophages

**DOI:** 10.3390/ijms21062254

**Published:** 2020-03-24

**Authors:** Emilia Metryka, Patrycja Kupnicka, Patrycja Kapczuk, Donata Simińska, Maciej Tarnowski, Marta Goschorska, Izabela Gutowska, Dariusz Chlubek, Irena Baranowska-Bosiacka

**Affiliations:** 1Department of Biochemistry and Medical Chemistry, Pomeranian Medical University in Szczecin, Powstańców Wlkp. 72 Str., 70-111 Szczecin, Poland; emilia_metryka@o2.pl (E.M.); patrycjakupnicka@o2.pl (P.K.); patrycja2510@o2.pl (P.K.); d.siminska391@gmail.com (D.S.); rcmarta@wp.pl (M.G.); dchlubek@pum.edu.pl (D.C.); 2Department of Physiology, Pomeranian Medical University in Szczecin, Powstańców Wlkp. 72 Str., 70-111 Szczecin, Poland; maciejt@sci.pum.edu.pl; 3Department of Medical Chemistry, Pomeranian Medical University in Szczecin, Powstańców Wlkp. 72, 70-111, Poland; izagut@pum.edu.pl

**Keywords:** lead (Pb), inflammation, cyclooxygenases, IL-1β, IL-6, THP-1 macrophages

## Abstract

The aim of this study was to assess the influence of lead (Pb) at low concentrations (imitating Pb levels in human blood in chronic environmental exposure to this metal) on interleukin 1β (IL-1β) and interleukin 6 (IL-6) concentrations and the activity and expression of COX-1 and COX-2 in THP-1 macrophages. Macrophages were cultured in vitro in the presence of Pb at concentrations of: 1.25 μg/dL; 2.5 μg/dL; 5 μg/dL; 10 μg/dL. The first two concentrations of Pb were selected on the basis of our earlier study, which showed that Pb concentration in whole blood (PbB) of young women living in the northern regions of Poland and in the cord blood of their newborn children was within this range (a dose imitating environmental exposure). Concentrations of 5 μg/dL and 10 μg/dL correspond to the previously permissible PbB concentrations in children or pregnant women, and adults. Our results indicate that even low concentrations of Pb cause an increase in production of inflammatory interleukins (IL-1β and IL-6), increases expression of COX-1 and COX-2, and increases thromboxane B2 and prostaglandin E2 concentration in macrophages. This clearly suggests that the development of inflammation is associated not only with COX-2 but also with COX-1, which, until recently, had only been attributed constitutive expression. It can be concluded that environmental Pb concentrations are able to activate the monocytes/macrophages similarly to the manner observed during inflammation.

## 1. Introduction

The ever-growing awareness of the harmful effects of lead (Pb) on the environment and human health has resulted in a reduction in the use of this metal in the production of fuels, paints, ceramics, and batteries [1]. Nevertheless, Pb is still considered to be one of the main substances posing the greatest potential threat to human health. In 2017, it was classified as second on the substance priority list created by the agency for toxic substances and disease registry (ATSDR) [2]. Chronic exposure to low Pb concentrations is a serious public health problem in large urban agglomerations and industrial areas [3,4,5]. Inhabitants of these areas are constantly exposed to low doses of Pb, which may lead to cognitive impairment during the development and disorders in neurobehavioral functioning, for example, aggression [6,7,8,9]. By directly influencing the expression of selected proteins and receptors and the activity of enzymes involved in neurodegenerative and neuroinflammatory processes, Pb may accelerate the progression and intensify the symptoms of Alzheimer′s disease (AD) and Parkinson′s disease (PD). The immune system seems to be one of the more sensitive targets for Pb. Although in low environmental concentrations, Pb does not cause visible damage to the body′s main immune cells, it may adversely affect their function [10,11,12,13,14].

Cyclooxygenase 1 (COX-1) and cyclooxygenase 2 (COX-2) are prostaglandin peroxide synthases, which play an important role in inflammation. The reactions they catalyze rely on the use of arachidonic acid (AA) as a substrate, obtained through the activity of cytoplasmic phospholipase A2 (cPLA2) from cell membrane phospholipids [15]. Both cyclooxygenase isoforms catalyze the transformation of AA into prostaglandin H2 (PGH2), from which the remaining prostaglandins and thromboxanes are synthesized [16,17,18]. The COX-1 gene expression in most cells is constitutive [19]. Until recently, COX-1 was considered not to be involved in the development of inflammation but recent works indicate that in some tissues this enzyme is also involved in the development of inflammation, both in the early stages of response to the proinflammatory factor [16] and during resolution of inflammation [20]. In turn, COX-2 is an enzyme subject to inducible expression in response to mitogens and proinflammatory cytokines produced by inflammatory cells, including monocytes [21]. COX-2 is considered to be the dominant source of prostaglandins in the ongoing inflammation, with prostaglandin E2 (PGE2) as its main product [16].

Macrophages are a very important element of a well-functioning immune system and it seems crucial to understand how their functions are affected by exposure to Pb, especially in light of the persistent contamination of the environment with heavy metals, and the increased incidence rate of inflammatory and autoimmune diseases in metropolitan populations. The aim of this study is to assess the influence of Pb at low concentrations (imitating Pb levels in human blood in chronic environmental exposure to this metal) on interleukin 1β (IL-1β) and interleukin 6 (IL-6) concentrations and the activity and expression of COX-1 and COX-2 in THP-1 macrophages.

## 2. Results

### 2.1. Cyclooxygenase 1

Each of the Pb concentrations applied after 24 h of incubation resulted in a statistically significant increase in messenger RNA (mRNA) COX-1 expression in relation to control. The highest applied concentration (10 μg/dL Pb) resulted in the strongest response to Pb (*p* = 0.04). The expression of COX-1 protein showed a similar upward trend as the results of mRNA expression, but statistical analysis did not show any significant differences, perhaps due to the low number of sample repetitions (*n* = 3).

Only the lowest of the applied Pb concentrations (1.25 μg/dL) resulted in no statistically significant increase in mRNA COX-1 expression after 48 h of exposure. Each of the higher concentrations induced increased expression of mRNA of the enzyme in relation to control. The highest expression was observed in the group of cells cultured with 5 μg/dL of Pb (*p* = 0.03). A significant increase in COX-1 protein expression was observed after incubation with the two highest Pb concentrations: 5 μg/dL (*p* = 0.02) and 10 μg/dL (*p* = 0.03). Cells exposed to 10 μg/dL of Pb showed the highest expression (Figure 1). 

### 2.2. Cyclooxygenase 2

The mRNA COX-2 expression after the 24-h incubation was highest in cells exposed to 2.5 μg/dL. This increase was significant in relation to the control group (*p* = 0.03) and to the highest of the applied concentrations (*p* = 0.03). The difference between the expression of mRNA at 10 μg/dL of Pb and control was also statistically significant (*p* = 0.03). Both of the highest Pb concentrations (5 μg/dL and 10 μg/dL) caused a significant increase in COX-2 protein expression vs. control (*p* = 0.03). The difference in expression between cells exposed to the highest and lowest of the tested Pb concentrations was also statistically significant (*p* = 0.03).

The 48-h incubation resulted in increased expression of mRNA COX-2 in cells cultured with Pb at 2.5 μg/dL, 5 μg/dL and 10 μg/dL vs. control (*p* = 0.03). The strongest influence on the expression of COX-2 protein was exerted by Pb at the highest studied concentration. The differences obtained between 10 μg/dL of Pb and other concentrations and controls were statistically significant at *p* = 0.03 (Figure 2).

### 2.3. Thromboxane A2 and Prostaglandin E2

The two highest concentrations seem to had strong effect on PGE2 production in macrophages after 24-h incubation, but only the growth at 5 μg/dL of Pb was significant relative to 2.5 μg/dL of Pb (*p* = 0.03). The increase in PGE2 secretion by cells cultured in 5 μg/dL of Pb for 48 h was significant with respect to each of the other concentrations and control group (*p* = 0.04) (Figure 3). Pb at 5 μg/dL caused the strongest increase in thromboxane B2 (TXB2) concentration in the studied macrophages after 24-h of exposure (*p* = 0.04). Moreover, the 48 h incubation resulted in a significant increase in TXB2 production due to 5 μg/dL of Pb. The increase was significant vs. control and the lowest of the applied Pb concentrations (1.25 μg/dL) (*p* = 0.03) (Figure 3). 

### 2.4. Interleukin 1β and Interleukin 6

Pb at 10 μg/dL caused the strongest increase in IL-1β concentration for both tested incubation times. The result obtained after 24 h was significant for the control group (*p* = 0.04) and cultured with 1.25 μg/dL Pb (*p* = 0.03). The increase in IL-1β concentration in the 48-h incubated cell culture medium with 10 μg/dL Pb was significant for 1.25 μg/dL Pb, 2.5 μg/dL Pb and control (*p* = 0.04) (Figure 4). Moreover, 24 h incubation of macrophages with the two highest Pb concentrations resulted in a significant increase in IL-6 concentration vs. 2.5 μg/dL Pb. Significant changes in IL-6 concentrations at 48-h exposure were found between the control and 2.5 μg/dL (*p* = 0.04) and 10 μg/dL of Pb (*p* = 0.04) (Figure 4). 

## 3. Discussion

In our study, we present the effect of environmental Pb concentrations on the expression of proinflammatory enzymes (COX-1 and COX-2), their products (PGE2 and TXB2), and interleukins (IL-1β and IL-6) in THP-1 macrophages at two different incubation times. 

At the cellular level, Pb is considered to be one of the factors inducing inflammation in the brain [11]. The results of our previous work on the expression of selected proteins, enzyme activity, and expression of receptors involved in neurodegenerative processes also show the direct role of Pb in the development of neurodegenerative changes in the brain [12,13,14]. In this study, Pb concentrations at both incubation times resulted in an increase in the mRNA and protein expression of COX-1 and COX-2. After 24-h incubation, the strongest expression of COX-1 mRNA was induced by the highest Pb concentration (10 μg/dL), while the highest expression of COX-2 mRNA was obtained for 2.5 μg/dL of Pb. Significantly, higher concentrations of Pb seemed to inhibit COX-2 mRNA expression. After incubation for 48 h, both enzymes showed the strongest mRNA expression at 5 μg/dL of Pb. The COX-1 gene is principally homeostatic in function and possesses a typical, GC-rich housekeeping promoter. In contrast, the COX-2 gene resembles an early response gene. It is strongly induced by mitogenic and proinflammatory stimuli, super induced by inhibitors of protein synthesis, and acutely regulated at both transcriptional and posttranscriptional levels [15,22]. Pb has been shown to increase mRNA levels in early *Fos* and *Jun* genes [23,24] which may explain the observed differences in mRNA expression of the studied cyclooxygenases in response to Pb. Observed changes in cyclooxygenases expression suggesting also the existence of some posttranscriptional, translational, and posttranslational events that result in silencing of those genes’ expression.

In our study, 24-h incubation did not significantly affect the protein expression of COX-1 perhaps due to the low number of sample repetitions in our study. However, longer exposure to Pb resulted in a significant increase in protein expression in cultures with higher Pb concentrations (5 μg/dL and 10 μg/dL). The increase in COX-2 protein expression was dose-dependent after 24 h incubation. This effect was not observed in a 48-h incubation. The mRNA and protein expression of the enzymes tested was not the same in all cases, which is not unique. For example, Gry et al. (2007) studied this phenomenon by evaluating 1066 gene products in 23 human cell lines and found significant correlations only in one third of examined mRNA species and corresponding proteins [25].

Our results are similar to those described by other researchers. Wei et al. (2014) tested the effect of Pb at concentrations from 25 μM to 100 μM on COX-2 induction in different cell types: rat C6 glioma cells, mouse BV2 microglia, in primary cultures of cortex neurons, in neural stem cells (NSCs) and RBE4 cells (brain endothelium) [26]. Increased expression of COX-2 was recorded in C6, BV2, primary culture of cortical neurons and NSCs [27]. In RBE4 cells, on the other hand, Pb caused only a slight increase in COX-2 gene expression only at doses greater than 50 μM [26]. In the human epidermoid carcinoma cell line A431, exposure to 1 µM of Pb(NO_3_)_2_ for 0.5 h, 1 h, or 2 h resulted in a time-dependent increase in both mRNA and protein expression of COX-2 [28]. Simões et al. (2015) studied the effect of low concentrations of Pb on the expression of COX-1 and COX-2, assuming that chronic treatment with low concentrations of lead should also increase oxidative stress and prostanoid pathways. They were breeding vascular smooth muscle cells (VSMCs) at 20 μg/dL of Pb for 48 h. Their results also indicate an increase in both mRNA and protein expression of COX-2. However, the researchers did not notice any changes in COX-1 expression [29]. There are many studies confirming the contribution of COX-1 and COX-2 to the development of neurodegenerative diseases such as Alzheimer′s disease [30,31,32,33,34]. COX-1 is known to actively participate in CNS immunoregulation [35,36,37,38]. Its increased expression have been shown in the microglia cells located around amyloid plaques in the post-mortem brains of patients with AD [39,40]. Activated microglia secrete an array of pro-inflammatory factors, such as prostaglandins, whose accumulation contributes to neuronal damages [41,42,43]. Alzheimer′s disease patients′ brains also exhibit an increased expression of COX-2 [44,45,46], which through its effect on total amyloid precursor proteins (APP) is correlated with the severity of brain amyloid plaque pathology [44]. In our experiment, low Pb concentrations (2.5μg/dL) resulted in a significant increase in the expression of COX-1 mRNA. Therefore, we can conclude that there is an increased expression of both cyclooxygenases in THP-1 cells exposed to Pb. This clearly suggests that the development of inflammation is associated not only with COX-2 but also with COX-1, which, until recently, had only been attributed constitutive expression.

Transcriptional regulation of the COX-2 gene is very complex because it can cover many signaling pathways and the mechanism changes depending on the stimulus and cell type. Simões and others (2015) demonstrated the contribution of p38 and p42/44 MAPK to the time-dependent increased expression of mRNA and COX-2 protein in response to Pb [29]. It is believed that increased expression of mRNA and COX-2 proteins in THP-1 cells may be achieved by activation of nuclear factor-kappaB (NF-kappaB), protein kinase C (PKC), p38 mitogen-activated protein kinase (p38 MAPK), NF-kappaB superrepressor and cAMP-responsive element binding (CREB) pathways [47,48]. 

Interleukins 1β and 6 are mainly synthesized by monocytes and macrophages [49]. Exposure to Pb also increases the concentration of arachidonic acid in phospholipid fractions of macrophages [50,51], which may contribute to the increase of PGE2 and TXB2 concentrations.

Our results indicate that even low concentrations of Pb can influence on production of inflammatory interleukins (IL-1β and IL-6) in THP-1 macrophages. The highest increase was observed in cultures exposed to the highest concentration used in our experiment, i.e., 10 μg/dL of Pb. In addition, the longer incubation period intensified cytokine production. In our research, we also observed a decrease in IL-6 concentration over 60% at a Pb concentration of 2.5 μg / dL vs. control, while higher Pb concentrations caused an increase in IL-6 concentration by over 50% vs. control. Although the observed changes were not statistically significant vs. control, they may indicate inhibition of IL-6 synthesis by Pb already at very low concentrations, and disturbances in the mechanisms controlling the synthesis of this protein by affecting the mechanisms, regulated at both transcriptional and posttranscriptional levels. However, this requires further research. Other researchers also observed a Pb-induced increase in the expression of proinflammatory interleukins and prostanoids. Flohé et al. (2002) studied the potential effect of lead chloride (PbCl_2_) on the release of cytokines and other inflammatory mediators by bone marrow macrophages (BMMφ) taken from young female mice. Authors demonstrated a time-dependent increase in the secretion of IL-6 and PGE2 [52]. An increased production of PGE2 in mouse marrow cell cultures in response to Pb has also been reported by Miyahara et al. (1994) [53].

In our study, we observed that Pb in low environmental concentrations is able to active the macrophages similarly to the manner observed during inflammation.

## 4. Materials and Methods 

### 4.1. Materials 

The cultured THP-1 cells came from the American Type Culture Collection (ATCC, Rockville, USA). The medium Roswell Park Memorial Institute (RPMI) 1640 and phosphate buffered saline (PBS) were purchased from Biomed Lublin (BIOMED, Lublin, Poland). Antibiotics (penicillin and streptomycin), needed for breeding, came from Sigma-Aldrich (Poznań, Poland) and fetal bovine serum (FBS) from Gibco (Paisley, UK). Lead acetate (PbAc) used for the preparation of solutions at specific concentrations of the tested substances came from Sigma-Aldrich (Poznań, Poland). Phorbol myristate acetate (PMA), needed to convert monocytes into macrophages, was purchased from Sigma-Aldrich (Poznań, Poland). Total RNA was extracted from the cells using Qiagen′s AllPrep DNA/RNA Mini Kit. Reverse transcription was performed using the high capacity complementary DNA (cDNA) Reverse Transcription Kit (Life Technologies, Carlsbad, CA, USA). Prevalidated Taqman Gene Expression Assays (Applied Biosystems, Waltham, MA, USA) and TaqMan Gene Expression (GE) Master Mix (Life Technologies, Carlsbad, CA, USA) were also used. The Power SYBR Green PCR Master Mix (Thermofisher) was used to quantify mRNA levels. In the Western Blot analysis, a radioimmunoprecipitation assay (RIPA) buffer (pH 7.4) containing: 20 mM Tris; 0.25 mM NaCl; 11 mM EDTA; 0.5% NP-40, 50 mM sodium fluoride and protease, Tris (Sigma-Aldrich, Poland), Tween 20 (Sigma-Aldrich, Poland), primary antibodies COX-1 (sc-19998, Santa Cruz Biotechnology, USA), COX-2 (ab 62331, Abcam, Cambridge, United Kingdom), anti-ß-actin (sc-47778, Santa Cruz Biotechnology, USA), secondary antibodies for detected COX-1 and ß-actin (anti-mouse, sc-516102, Santa Cruz Biotechnology, USA), COX-2 (anti-rabbit, ab 97051, Abcam, Cambridge, United Kingdom), phosphatase inhibitors (Sigma, Poznań, Poland), which were used for cell homogenization and nitrocellulose membrane, which was used for transfer (Thermo Scientific, Pierce Biotechnology, USA). In order to visualize proteins, a solution for chemiluminescence excitation (Novex ECL Chemiluminescent Substrate Reagent Kit, Invitrogen) was used. PGE2 and TXA2 were extracted using Bakerbond SPE columns (J.T. Baker, Phillipsburg, NJ, USA). The measurements of PGE2 and TXB2 levels were conducted using appropriate immunoenzymatic sets (Prostaglandin E2 EIA Kit, Cayman, USA; Thromboxane B2 EIA Kit, Cayman, USA). The measurements of IL-1β and IL-6 levels were using specific Quantikine ELISA Kits by R&D Systems (Abingdon, UK).

### 4.2. THP-1 Macrophages Experimental Model

This study was carried out on leukemia cell line THP-1. These cells are mainly used in experiments investigating mechanisms of inflammatory reactions [54,55]. They are also commonly used as a research model in various works to study the physiology or pathology of human monocytes and macrophages [56,57,58,59,60,61].

THP-1 macrophages were cultured in vitro in the presence of Pb at concentrations of: 1.25 μg/dL; 2.5 μg/dL; 5 μg/dL; 10 μg/dL. The first two concentrations of Pb were selected on the basis of our earlier study [3], which showed that Pb concentration in whole blood (PbB) of young women living in the northern regions of Poland, and in the cord blood of their newborn children, was within this range (a dose imitating environmental exposure). Concentrations of 5 μg/dL and 10 μg/dL correspond to the previously permissible PbB concentrations in children or pregnant women, and adults [62]. The control group consisted of cells cultured in complete RPMI 1640 medium without addition of Pb or any irritant.

### 4.3. Cell Culture and Treatment

Cells were cultured in RPMI 1640 medium enriched with 10% FBS, 100 IU/mL penicillin, and 10 μg/mL streptomycin at 37 °C, 5% CO_2_ atmosphere and 95% humidity. Three times a week, cells were passaged to keep their density below 8 x 10^5^ cells/mL. The number of cells was determined using Burker′s chamber (Sigma-Aldrich, Poznań, Poland) with a light microscope. Cell viability was examined using a trypan blue dye exclusion method. Cell cultures with a viability of more than 97% were used for experiments [63]. During the experiment, in order to differentiate monocytes into macrophages, cells were incubated with 100 nM PMA for 24 h (Figure 5). The macrophages attached to the substrate were washed three times with warm PBS and were incubated with lead acetate (PbAc) for 24 and 48 h. After the incubation time, the cells were scraped with a cell scraper and transferred to the tubes. The material was then centrifuged (125×G for 6 min). After the centrifugation, the supernatant was poured into a separate tube. The cellular pellet was rinsed in 1 mL of PBS and re-centrifuged. After the removal of PBS, the obtained cell pellets and culture medium were frozen and stored at −80 °C until the test. Protein concentration was measured using the Bradford method [64].

### 4.4. Gene Expression

#### Cyclooxygenase-1 (COX-1) and Cyclooxygenase-2 (COX-2) Gene Expression Analysis by qRT-PCR

Quantitative expression of mRNA of PTGS1 (National Center for Biotechnology Information (NCBI) Reference Sequence: NM_000962) and PTGS2 (NM_000963) genes was performed in a two-stage reverse transcription PCR. Total RNA was isolated from cells with an AllPrep DNA/RNA Mini Kit. Concentrations and purity of obtained RNA were measured by the NanoDrop ND-1000 spectrophotometer (NanoDrop Technologies, Waltham, MA, USA). The RNA was reverse-transcribed using a high capacity cDNA Reverse Transcription Kit with random primers, according to the manufacturer’s instructions. Quantitative assessment of mRNA levels was performed by the 7500 Fast Real-Time PCR System (Applied Biosystems, Waltham, MA, USA), with Power SYBR Green PCR Master Mix reagent. Every sample was analyzed simultaneously in two technical replicates and mean cycle threshold (CT) values were used for further analysis. The relative quantification method was applied in calculations, using 7500 Fast Real-Time PCR System Software (Applied Biosystems, Foster City, CA, USA). The relative quantity of a target, normalized to the endogenous control GAPDH gene. Analysis of these relative changes in gene expression between samples was performed using the 2^−ΔΔ*C*T^ method [65].

### 4.5. Protein Expression

#### Western Blotting Analysis of COX-1 and COX-2 Expression

The cells obtained after incubation with the aforementioned lead concentrations were lysed with RIPA Lysis Buffer System for 2 h at fridge temperature with constant shaking. After standard SDS-PAGE separation, the proteins were transferred onto nitrocellulose membranes using wet transfer at 75 V for 1 h at room temperature. The membrane was blocked for one hour in 5% fat-free milk in Tris buffered saline containing 0.1% Tween 20 (TTBS). It was then incubated overnight in 4 °C with primary monoclonal COX-1 antibodies at a 1:200 dilution or COX-2 antibodies at a 1:1000 dilution with a monoclonal anti-Beta-actin (β-actin) (1:4000) and then for one hour at room temperature with secondary anti-mouse COX-1 or anti-rabbit antibodies for detected COX-2 at a 1:6000 dilution and ß-actin at a 1:3000 dilution. In order to visualize proteins, the membrane was incubated for 2 min in 4 mL of chemiluminescent solution and placed in a transiluminator (Molecular Imager ChemiDock XRS+, BIO-RAD, Poznań, Poland).

### 4.6. Determination of COX-1 and COX-2 Activity

The activity of COX-1 and COX-2 was measured in vitro by quantitative measurement of products catalyzed by these reaction enzymes: thromboxane A2 (TXA2) and PGE2. Before PGE2 and TXA2 determination, acetate buffer acidified to pH ~ 4.0 was extracted from the culture medium using Bakerbond SPE columns, according to the manufacturer′s instructions. Concentrations of PGE2 were tested spectrophotometrically by enzyme immunoassay kit PGE2. TXA2 is unstable (half-life: 37 sec) and is rapidly hydrolyzed non-enzymatically to TXB2 (its stable derivative), so the Thromboxane B2 Enzyme Immunoassay Kit was used to measure free TXA2 indirectly.

### 4.7. The Measurements of Interleukin 1β (IL-1β) and Interleukin 6 (IL-6) Concentration

The concentration of proinflammatory interleukins was examined in the culture medium of experimental cells. Determination was carried out using the specific Quantikine ELISA Kits by R&D Systems (Abingdon, UK) according to the instructions provided by the manufacturer.

### 4.8. Statistical Analysis

The obtained results were analyzed using the Statistica 10.0 software package. Results were expressed as mean and standard deviation (X ± SD). As most of the distributions deviated from the normal distribution (Shapiro–Wilk test), non-parametric tests were used for the analyses. Mann–Whitney test was used to assess the differences between the tested parameters. A probability at *p* ≤ 0.05 was considered statistically significant.

## Figures and Tables

**Figure 1 ijms-21-02254-f001:**
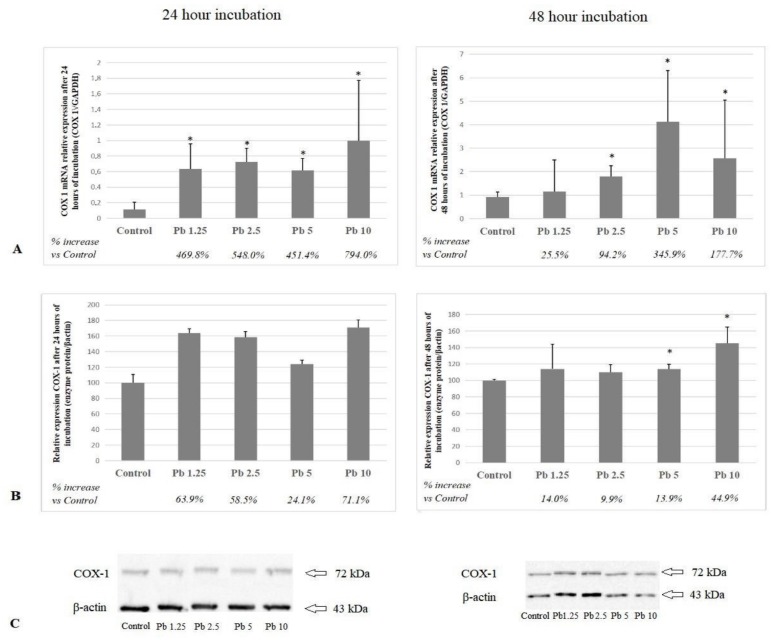
The effect of lead (Pb) on COX-1 messenger RNA (mRNA) and protein expression in macrophages after 24 and 48 h of incubation. (**A**) COX-1 mRNA expression following Pb exposure; (**B**) COX-1 protein expression (densitometric analysis) of protein normalized to Beta-actin (β-actin); (**C**) representative Western blot following Pb exposure. Macrophages were cultured with Pb solutions for 24 or 48 h. After incubation, cells were harvested by scraping and mRNA was measured using the real-time PCR method (*n* = 6) and protein expression by using the Western blotting method (*n* = 3). *Statistically significant differences vs. control (*p* ≤ 0.05). Control—cells incubated in Roswell Park Memorial Institute (RPMI) medium with 10% fetal bovine serum (FBS) and without Pb.

**Figure 2 ijms-21-02254-f002:**
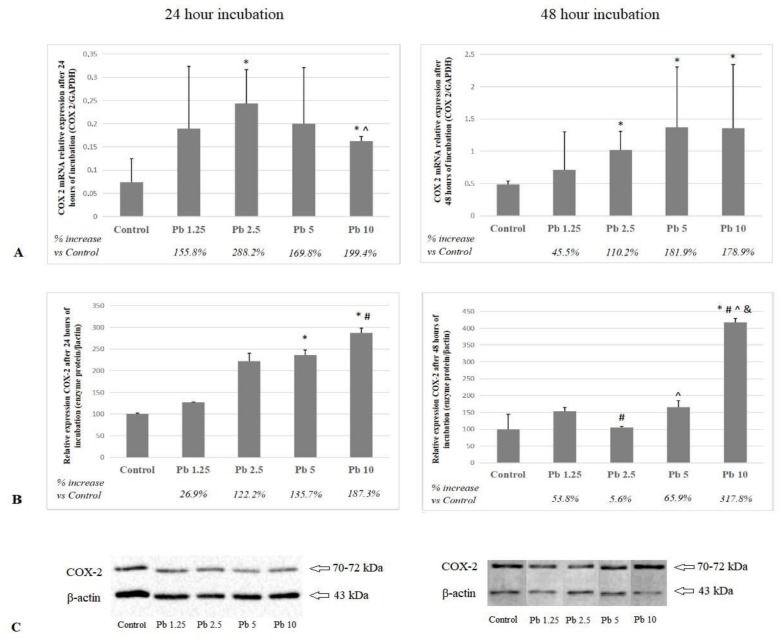
The effect of Pb on COX-2 mRNA and protein expression in macrophages after 24 and 48 h of incubation. (**A**) COX-2 mRNA expression following lead exposure; (**B**) COX-2 protein expression (densitometric analysis) of protein normalized to β-actin; (**C**) representative Western blot following lead exposure. Macrophages were cultured with lead solutions for 24 or 48 h. After incubation, cells were harvested by scraping and mRNA was measured by using the real-time PCR method (*n* = 6) and protein expression by using the Western blotting method (*n* = 3). * Statistically significant differences vs. control (*p* ≤ 0.05). # Statistically significant differences vs. 1.25 μg/dL (*p* ≤ 0.05). ^ Statistically significant differences vs. 2.5 μg/dL (*p* ≤ 0.05). & Statistically significant differences vs. 5 μg/dL (*p* ≤ 0.05). Control—cells incubated in RPMI medium with 10% FBS and without Pb.

**Figure 3 ijms-21-02254-f003:**
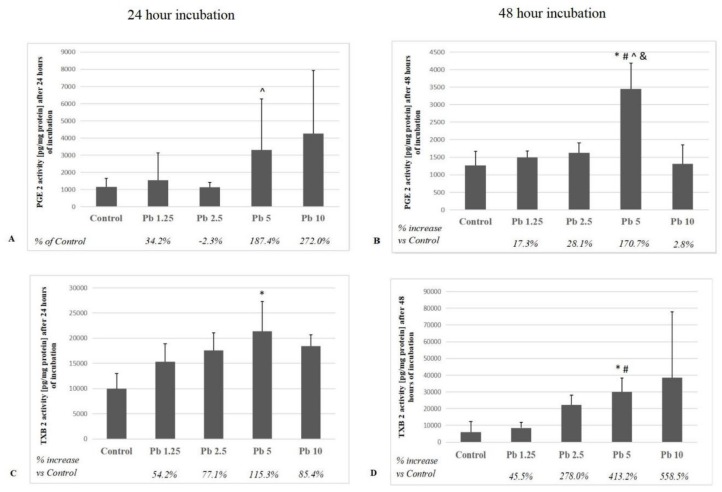
The effect of Pb on quantity of PGE2 and TXB2 in culture supernatants of macrophages cultured with various lead solutions for 24 h (**A**, **C**) or 48 h (**B**, **D**). Macrophages were cultured with lead solutions for 24 or 48 h. After incubation, cells were harvested by scraping and PGE2 or TXB2 and concentrations were measured by the ELISA method (*n* = 6). * Statistically significant differences vs. control (*p* ≤ 0.05). # Statistically significant differences vs. 1.25 μg/dL (*p* ≤ 0.05). ^ Statistically significant differences vs. 2.5 μg/dL (*p* ≤ 0.05). & Statistically significant differences vs. 10 μg/dL (*p* ≤ 0.05). Control—cells incubated in RPMI medium with 10% FBS and without Pb.

**Figure 4 ijms-21-02254-f004:**
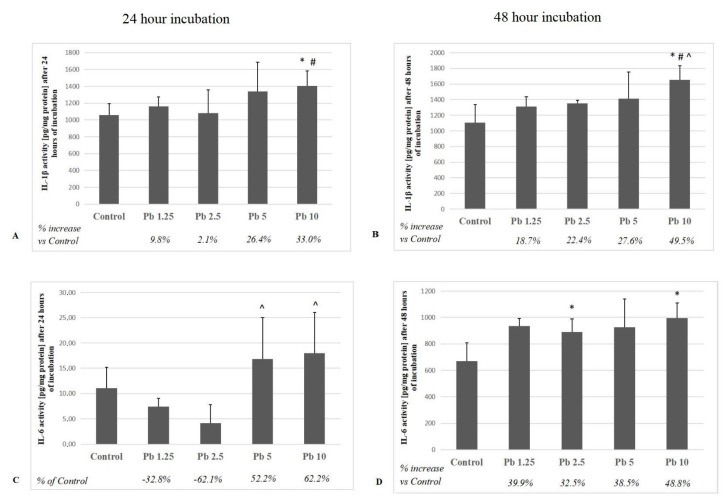
The effect of Pb on the quantity of IL-1β and IL-6 in culture supernatants of macrophages cultured with various Pb solutions for 24 h (**A**, **C**) or 48 h (**B**, **D**). Macrophages were cultured with lead solutions for 24 or 48 h. After incubation, cells were harvested by scraping and IL-1β or IL-6 concentration was measured by the ELISA method (*n* = 6). * Statistically significant differences vs. control (*p* ≤ 0.05). # Statistically significant differences vs. 1.25 μg/dL (*p* ≤ 0.05). ^ Statistically significant differences vs. 2.5 μg/dL (*p* ≤ 0.05). Control—cells incubated in RPMI medium with 10% FBS and without Pb.

**Figure 5 ijms-21-02254-f005:**
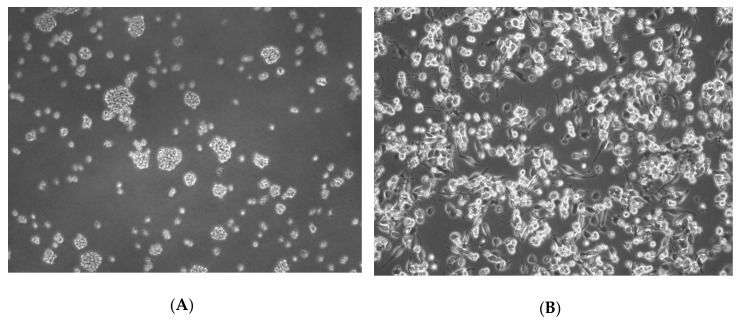
Culture of THP-1 (A) monocytes cultured in RPMI medium with 10% FBS; (B) macrophages after 24 h of incubation with phorbol myristate acetate (PMA). Normal cells are visible.

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
