# Peer review of "Lead (Pb) as a Factor Initiating and Potentiating Inflammation in Human THP-1 Macrophages"

_ijms, 2020, doi:10.3390/ijms21062254_

Round 1

Reviewer 1 Report

The study aims in assessing in vitro the pro-inflammatory action of low concentrations of Pb (as reported in human blood) on macrophages. Il1beta, IL6, as well as Cox1, Cox2 and their products (PGE2 and TBX2) have been analyzed after 24 and 48h of challenging. A tentative conclusion on the potential of Cox1 and microglia activation on neurodegeneration have been presented.

Despite the topic might be interesting, especially in respect to Cox1 induction by Pb (the unique novelty of the work), the work present several problems.

It is difficult understanding the real goal of the work. Is it the evaluation of Cox1 role under Pb challenging (the real novelty)? Is it the microglia/brain activation by Pb? or blood macrophages activation by Pb? The experimental setting is cofounding. THP-1 macrophages have been used and challenged with concentration of Pb noticed in blood of human subjects, but in the paper, microglia and Pb effect on brain seems to be the target. This looks not correct. If Pb toxicity to brain (to microglia) is the goal of the work, microglia (cell lines or primary cultures) have to be used and challenged with concentration of lead noticed in the CNS. Data to mimic the CNS environment (brain Pb concentrations) might be found in experiments using animal models and assessing both blood as well as CNS Pb concentrations (As example, your previous work doi: 10.1016/j.tox.2012.10.10.027, but additional references may be found in PubMed). Information on Pb levels in human brain may be found, as example, in Grochowski, 2019, Frontiers in chemistry. DOI: 10.3389/fchem.2019.00115. If the topic is Pb effect on blood macrophages, or Cox1 modulation, the discussion and even more importantly, the conclusions (abstract and main text) have to be limited to blood/macrophages and extrapolation to brain/microglia clearly highlighted as speculative or even better avoided. Modulation of Cox1 in brain inflammation (in non Pb linked models) is known (as example doi: 10.1093/jnen/nlx056; doi: 10.3389/fneur.2017.00251) and its specific contribution has been elucidated by using specific inhibitors (as example: doi: 10.3389/fneur.2017.00251; doi: 10.1186/s12974-017-1014-3). The cited papers might be useful for improving the work by dissecting the contribution of Cox1 and Cox2 in the specific field of Pb poisoning. This could be the real novelty. Cytokines as well PGE2 and TBX2 (protein level - figure 3 and 4) have been analyzed uniquely in the cell lysate. It is important to analyze also the level in the supernatant. This might change the results and thus the interpretation of the data concerning the induction due to the different concentrations of Pb, as well as the different timing (24 vs. 48h). Regarding Figure 1, interesting is the Cox1 protein increase at 24h (with a very good SD), that is ever higher that the increase at 48h. Thus, it looks strange that there is not statistical significance at 24h of challenging. Increasing the number of biological repetitions (actually 3) analyzed by Western blot will surely resolve the problem, changing the meaning of the data and of the whole work, so is required. Figure3 and 4: please add the indication oh the timing (24 – 48 hour of incubation) as in figs 1 and 2. Result section appears frequently repetitive. It has to be written again in a more concise way, improving the understanding of the content. Discussion, sentence 1 (lines: 190-193). Please moderate the sentence, despite it might be true that the effect of Pb on cytokines, Cox and their products have been evaluated for the first time in THP-1 cells, the general findings reported by the papers are not so new. Please avoid sensationalisms. Discussion. In general, the interpretation of the results described in the discussion is missing, the general organization of the section is weak, and the relevant information contained in the cited literature are not used for discussing the data of the present work. Please rearrange following the suggestions here given. Lines: 203-206. it is true that the maximal induction of Cox2 was reached with Pb 2.5, while Cox1 maximal modulation was at Pb10, but Cox1 was ever several time more induced (from 470 at 1.25Pb to 800% at 10Pb vs. control) than Cox2 (from 150% at 1.25 Pb to 200% at 10pb vs control). Because Cox1 has been considered constitutive for long time (as you discussed), this is interesting. Please give you interpretation of the results in the discussion. Discussion, lines 206-207. At longer exposition (48h), both Coxs mRNA modulation is weaker than at 24h. To me, this behavior looks like a response to an acute/high doses stimuli. See also comment #7 before commenting in the discussion. Discussion, lines: 208-210. See point #7. Discussion, lines 216-236, plus 237-243. Cox2 modulation is well known, but also Cox1 induction has been already studied in different CNS pathological conditions/models (see point #4). Notably, the individual contribution of Cox1 (vs Cox2) has been unraveled by using specific inhibitors. Thus, the simple observation that in your model both Coxs are modulated is insufficient to affirm that Cox 1 is relevant in triggering the inflammatory cascade under Pb challenging (see point #4). Second, additional discussion on what is known about Cox1 modulation and its role in inflammation in different models/cells/organs (similarly to what has been discussed for Cox2) has to be added. Thus, please expand, and better state the most relevant findings about Cox1 in neuroinflammation, contained in the cited literature from line 237-243. Discussion, lines 244-271. It is not clear the usefulness of this information in the context of the present work. Please write again in a more concise and focused way, or it is suggested to remove. A shorter but more focused discussion may improve the readability of the Ms. Discussion, lines 291-296. The description is not well in agreement with the data in fig 4, where, IL1beta is increased at both timing only under the maximal challenging with Pb (10 Pb), while IL6 require the two higher concentration of Pb at 24h, while a significant increase is present at 2.5 at 48h. Please see also point #6, and #2, before commenting.

Author Response

It is difficult understanding the real goal of the work. Is it the evaluation of Cox1 role under Pb challenging (the real novelty)? Is it the microglia/brain activation by Pb? or blood macrophages activation by Pb?

The experimental setting is cofounding. THP-1 macrophages have been used and challenged with concentration of Pb noticed in blood of human subjects, but in the paper, microglia and Pb effect on brain seems to be the target. This looks not correct. If Pb toxicity to brain (to microglia) is the goal of the work, microglia (cell lines or primary cultures) have to be used and challenged with concentration of lead noticed in the CNS. Data to mimic the CNS environment (brain Pb concentrations) might be found in experiments using animal models and assessing both blood as well as CNS Pb concentrations (As example, your previous work doi: 10.1016/j.tox.2012.10.10.027, but additional references may be found in PubMed).

The aim of our paper was clearly to determine “the influence of Pb at low concentrations (imitating Pb levels in human blood in chronic environmental exposure to this metal) on interleukin 1β (IL-1β) and interleukin 6 (IL-6) concentrations and the activity and expression of COX-1 and COX-2 in THP-1 macrophages”.

It was not our aim to investigate microglia (as Reviewer suggests). We only indicated in the Introduction that the cells we used to conduct this research are a good and commonly used model in the research on the effect of various factors on the metabolism of these cells, and therefore the brain function and the immune system.

The cells we have used as the research model also show a number of common features with peripheral blood macrophages and microglia, e.g. expression of CD68, a well-known marker of THP-1 macrophages. Reviewer suggests we should use completely different cells for our research, i.e. microglia cell lines or primary cultures, which would mean creating a completely different research model and performing totally different and new experiments, something that is currently impossible for us. According to Reviewer remark we removed this fragment misleading fragment according to microglia from Introduction.

Information on Pb levels in human brain may be found, as example, in Grochowski, 2019, Frontiers in chemistry. DOI: 10.3389/fchem.2019.00115. If the topic is Pb effect on blood macrophages, or Cox1 modulation, the discussion and even more importantly, the conclusions (abstract and main text) have to be limited to blood/macrophages and extrapolation to brain/microglia clearly highlighted as speculative or even better avoided.

The parameter we use is blood Pb level. This is the widely used criterion of human exposure to Pb, used in establishing the permissible levels in people. By no means is it the brain Pb level, as Reviewer suggests. In our paper, THP-1 macrophages were cultured in vitro in the presence of Pb at concentrations of: 1.25 μg/dL; 2.5 μg/dL; 5 μg/dL; 10 μg/dL. The first two concentrations of Pb were selected on the basis of our earlier study, which showed that Pb concentration in whole blood of young women living in the northern regions of Poland and in the cord blood of their newborn children was within this range (a dose imitating environmental exposure). Concentrations of 5 μg/dL and 10 μg/dL correspond to the previously permissible PbB concentrations in children or pregnant women, and adults.

Modulation of Cox1 in brain inflammation (in non Pb linked models) is known (as example doi: 10.1093/jnen/nlx056; doi: 10.3389/fneur.2017.00251) and its specific contribution has been elucidated by using specific inhibitors (as example: doi: 10.3389/fneur.2017.00251; doi: 10.1186/s12974-017-1014-3). The cited papers might be useful for improving the work by dissecting the contribution of Cox1 and Cox2 in the specific field of Pb poisoning. This could be the real novelty.

Thank you for this comment, we have included the results of the following papers in the discussion:

Kaur C, Rathnasamy G, Ling EA. Biology of Microglia in the Developing Brain. J Neuropathol Exp Neurol. 2017 Sep 1;76(9):736-753. doi: 10.1093/jnen/nlx056.

Calvello R, Lofrumento DD, Perrone MG, Cianciulli A1, Salvatore R, Vitale P, De Nuccio F, Giannotti L, Nicolardi G, Panaro MA, Scilimati A. Highly Selective Cyclooxygenase-1 Inhibitors P6 and Mofezolac Counteract Inflammatory State both In Vitro and In Vivo Models of Neuroinflammation. Front Neurol. 2017 Jun 9;8:251. doi: 10.3389/fneur.2017.00251. eCollection 2017.

Saliba SW, Marcotegui AR, Fortwängler E, Ditrich J, Perazzo JC, Muñoz E, de Oliveira ACP, Fiebich BL AM, paracetamol metabolite, prevents prostaglandin synthesis in activated microglia by inhibiting COX activity. J Neuroinflammation. 2017 Dec 13;14(1):246. doi: 10.1186/s12974-017-1014-3.

Cytokines as well PGE2 and TBX2 (protein level - figure 3 and 4) have been analyzed uniquely in the cell lysate. It is important to analyze also the level in the supernatant. This might change the results and thus the interpretation of the data concerning the induction due to the different concentrations of Pb, as well as the different timing (24 vs. 48h).

The determinations of PGE2 and TXB2 were made in supernatant, although PCR and WB determinations were made in cell lysate, as indicated in the text: “Before PGE2 and TXA2 determination, acetate buffer acidified to pH ~ 4.0 was extracted from the culture medium using Bakerbond SPE columns, according to the manufacturer's instructions”

Regarding Figure 1, interesting is the Cox1 protein increase at 24h (with a very good SD), that is ever higher that the increase at 48h. Thus, it looks strange that there is not statistical significance at 24h of challenging. Increasing the number of biological repetitions (actually 3) analyzed by Western blot will surely resolve the problem, changing the meaning of the data and of the whole work, so is required.

Thank you for the attention of the reviewer, unfortunately we cannot currently perform another WB test for financial reasons.

Figure3 and 4: please add the indication oh the timing (24 – 48 hour of incubation) as in figs 1 and 2.

Thank you very much, we have corrected the figures.

Result section appears frequently repetitive. It has to be written again in a more concise way, improving the understanding of the content.

Thank you for your attention. The description of the results has been shortened.

Discussion, sentence 1 (lines: 190-193). Please moderate the sentence, despite it might be true that the effect of Pb on cytokines, Cox and their products have been evaluated for the first time in THP-1 cells, the general findings reported by the papers are not so new. Please avoid sensationalisms.

The sentence has been corrected according to Reviewer's comment.

Discussion. In general, the interpretation of the results described in the discussion is missing, the general organization of the section is weak, and the relevant information contained in the cited literature are not used for discussing the data of the present work. Please rearrange following the suggestions here given. Lines: 203-206. it is true that the maximal induction of Cox2 was reached with Pb 2.5, while Cox1 maximal modulation was at Pb10, but Cox1 was ever several time more induced (from 470 at 1.25Pb to 800% at 10Pb vs. control) than Cox2 (from 150% at 1.25 Pb to 200% at 10pb vs control). Because Cox1 has been considered constitutive for long time (as you discussed), this is interesting. Please give you interpretation of the results in the discussion. Discussion, lines 206-207. At longer exposition (48h), both Coxs mRNA modulation is weaker than at 24h. To me, this behavior looks like a response to an acute/high doses stimuli. See also comment #7 before commenting in the discussion. Discussion, lines: 208-210. See point #7. Discussion, lines 216-236, plus 237-243. Cox2 modulation is well known, but also Cox1 induction has been already studied in different CNS pathological conditions/models (see point #4). Notably, the individual contribution of Cox1 (vs Cox2) has been unraveled by using specific inhibitors. Thus, the simple observation that in your model both Coxs are modulated is insufficient to affirm that Cox 1 is relevant in triggering the inflammatory cascade under Pb challenging (see point #4).

According with the Reviewer's suggestion, we have added the explanation why COX-1 was subject to stronger modulation.

Second, additional discussion on what is known about Cox1 modulation and its role in inflammation in different models/cells/organs (similarly to what has been discussed for Cox2) has to be added. Thus, please expand, and better state the most relevant findings about Cox1 in neuroinflammation, contained in the cited literature from line 237-243.

According with the Reviewer's suggestion, we have included relevant findings about Cox1 in neuroinflammation.

Discussion, lines 244-271. It is not clear the usefulness of this information in the context of the present work. Please write again in a more concise and focused way, or it is suggested to remove.

A shorter but more focused discussion may improve the readability of the Ms. Discussion, lines 291-296.

According to the Reviewer's suggestion, we have shortened discussion.

The description is not well in agreement with the data in fig 4, where, IL1beta is increased at both timing only under the maximal challenging with Pb (10 Pb), while IL6 require the two higher concentration of Pb at 24h, while a significant increase is present at 2.5 at 48h. Please see also point #6, and #2, before commenting.

A comment was added as suggested by the reviewer.

Reviewer 2 Report

This is essentially a well- written paper on a well-designed experiment. The results are sound and the conclusions firmly based.

The MS could even be published as it is now, but certain modifications would improve its quality.

Style and grammar are free of major errors. However, some sentences are less easy to follow and could be rewritten for more clarity and simplicity.

Introduction:

Line 36: “batteries and accumulators” – these are the same, are they not?

L. 40-42: Complicated sentence.

L. 70: “myogens” – what did you mean?

Results:

Check if text description of the results and the corresponding graphs in the figures are in complete accordance. No need to give percent data in the text AND in the figures. No need to mention all changes and their significance, emphasize the main ones which show the general trend of the effects.

L. 108: Use “vs.” instead of “in comparison to…”

L. 109: Your “irritant” is Pb. Call it by its name.

L. 113: “highest of the applied concentrations (p=0.03)” – Significant vs. what?

L. 122-126 and Fig. 2: Again, even if you mark every significant difference in the graphs, mention only the important ones in the text.

Discussion:

Discussion is a bit lengthy, and the references are a bit too numerous. Check this chapter and delete statements that are only loosely connected to the main message.

Materials and Methods:

Section 3.3. describing your model could be placed before the section on cell culturing (now 3.2.).

Author Response

Comments and Suggestions for Authors

This is essentially a well- written paper on a well-designed experiment. The results are sound and the conclusions firmly based.

The MS could even be published as it is now, but certain modifications would improve its quality.

Style and grammar are free of major errors. However, some sentences are less easy to follow and could be rewritten for more clarity and simplicity.

We have corrected grammar and attached Certificate of Native Speaker

Introduction:

Line 36: “batteries and accumulators” – these are the same, are they not?

  1. 40-42: Complicated sentence.
  2. 70: “myogens” – what did you mean?

According to the Reviewer's suggestion, we corrected the sentence

Results:

Check if text description of the results and the corresponding graphs in the figures are in complete accordance. No need to give percent data in the text AND in the figures. No need to mention all changes and their significance, emphasize the main ones which show the general trend of the effects.

According to the Reviewer's remark we have corrected the graphs and figures.

  1. 108: Use “vs.” instead of “in comparison to…”
  2. 109: Your “irritant” is Pb. Call it by its name.
  3. 113: “highest of the applied concentrations (p=0.03)” – Significant vs. what?
  4. 122-126 and Fig. 2: Again, even if you mark every significant difference in the graphs, mention only the important ones in the text.

According to the Reviewer's remark we have corrected all mistakes.

Discussion:

Discussion is a bit lengthy, and the references are a bit too numerous. Check this chapter and delete statements that are only loosely connected to the main message.

According to the Reviewer's comment we have corrected discussion.

Materials and Methods:

Section 3.3. describing your model could be placed before the section on cell culturing (now 3.2.).

According to the Reviewer's remark, we have corrected this section.

Round 2

Reviewer 1 Report

Please see the Attcahed pdf

Author Response

New comment to points 1 and 2)

Response 1and 2 clarify that the experimental setting is reproducing the toxicity of blood levels of Pd on circulating macrophages. The introduction has been improved by removing confounding references to microglia and neurodegenerative disorders. Nevertheless, sentences and more importantly conclusion trying to extrapolating the results of the present work (blood Pb concentration on circulating macrophages) in the neuro-inflammatory and neuro degenerative context are still present, and look inappropriate as there are.

All references to neurodegeneration or neuroinflammation were removed from the text and conclusion as suggested by Reviewer.

New comment

Despite I can understand, the meaning of the data are really relevant. Indeed several

conclusion all long the discussion use the information as “well performed” and “final”, what is not true. Having confirmed the significance of the early induction (that actually is not reached only due to lack of few repetitions) will totally change the interpretation of the work. We cannot believe in the results as they are, thus additional biological repetitions have to be added, or clearly stated in the text that the data cannot be used.

According to Reviewer remark , it was indicated in the text that the results obtained may be the result of using too few repetitions.

New comments to the V2.

We believe that only marginal improvement have been done. Several critical points remains. Especially relevant is the use (and description) of the results, for which the interpretation as well as

the statistics looks erroneous, misleading and even enforced to support the desired result. Here, in the order of appearance in the Ms.

71-73: The highest applied concentration (10 μg/dL Pb) resulted in the strongest response to Pb (p=0.04). The highest expression was observed in the group of cells cultured with 5 μg/dL of Pb (p=0.03). I agree with the first one, but not with the “highest expression with 5 μg/dL of Pb”.

We notice, that when we editing of the previous version of manuscript, there was a shift in the text. The entire paragraph regarding COX-1 protein expression has disappeared. The sentence “The highest expression was observed in the group of cells cultured with 5 μg/dL of Pb (p=0.03)” it was already referring to Western Blott results, so it could have been misleading in interpretation. The sentence has been corrected.

73-75: A significant increase in COX-1 protein expression was observed after incubation with the two highest Pb concentrations: 5 μg/dL (p=0.02) and 10 μg/dL (p=0.03). The statistical significance is not indicated in the figure.

As above, text shift. We corrected the sentence.

77-79: “The comparison of results obtained at both incubation times showed that longer exposure to Pb resulted in a stronger expression of mRNA and COX-1 protein in each of the study groups”. Not true: The increase of both mRNA and Protein levels were higher at 24h (vs the respective control), rather than at 48h (vs the respective controls), as clearly stated by the % at the bottom of each column.

The statistics described here refer to comparisons of the results of both incubation times (24 and 48 hours together in each of the cell groups). Calculations were made for each of the parameters determined. These data are not placed on any of the graphs presented. In order not to disturb the interpretation, these data were completely removed (for all tested substances) as suggested by the Reviewer. Only statistics describing the significance between the examined groups in each of the incubation times were left in the text.

78—80: “A 48 hour incubation resulted in a significant increase in mRNA COX-1 in cultures with 2.5 μg/dL and 5 μg/dL of Pb (p=0.03)”. The description of the 10 Pp is lacking. In Figure 1 this concentration has been also indicated as significant.

As above, we corrected the sentence.

80-81: “Each of the concentrations used resulted in a significant increase in COX-1 expression after 48 hours compared to the 24-hour incubation (p≤0.05)”. Not true or unclear. The mRNA expression at 24h is about 550 (2.5ug/dl) and 450 (5ug/dL) fold vs. control, while at 48 is respectively 100 and 350fold vs. control, so lower. If the Author refers to the protein level, the text is not describing the content of Figure 1, because (based on Figure 1) Cox1 protein level at 48h is significantly modulated at 5 and 10ug/dL, not 2.5 and 5 ug/dL. Indeed, if the results at 48h have been compared with the data at 24h, this has to be indicated in the legend. This information is not present in the legend to Figure 1.

As above, we corrected the sentence.

104-105. “In cells exposed to Pb for 48h expressions of both mRNA and COX-2 proteins were significantly higher”. Unclear, please specify vs. what.

As above, we corrected the sentence.

105-106: “The 48 hour incubation period resulted in a significant increase in protein expression for each of the tested Pb concentrations (p=0.03)”. Based on Figure 2, this is not true: 1.25mg/dL do not leads to significant changes. Indeed, only the 10ug/dL results statistically significant vs. controls.

As above, we corrected the sentence.

119-120: “The two highest concentrations had a significantly stronger effect on PGE2 production in macrophages after 24-hour incubation”. Not true, significant relevance vs controls is never reported, and the unique significance is obtained comparing Pb 5ug/dL vs Pb 2.5 (lower than the control).

The sentence was corrected: “The two highest concentrations seem to had had a significantly stronger effect on PGE2 production in macrophages after 24-hour incubation, but only the growth at 5 μg / dL of Pb was significant relative to 2.5 μg / dL of Pb (p = 0.03). "

142-143: “24 h incubation of macrophages with the two highest Pb concentrations resulted in a significant 142 increase in IL-6 concentration”. Again, not significantly relevant vs control, which is of primary importance. The relevance vs 2.5 it is due only to the decreased level (-60%) of Il6 under 2.5ug/dl challenging. Explain this decrease.

It has been corrected that this difference is statistically significant for cells cultured with Pb at a concentration of 2.5ug / dl. An explanation for the observed decrease in IL-6 concentration has been added.

163-165 “In this study Pb concentrations at both incubation times resulted in an increase in the mRNA and protein expression of COX-1 and COX-2, with a longer incubation time (48-hour) resulting in a stronger cellular reaction”. By looking figure1 and 2 (details on the % of induction under each of the treatments), the sentence is not true. Both Cox1 and Cox2 mRNA and protein expression are higher (vs the respective

controls) at short times: 24h (exception are: Cox2 mRNA under 5ug/dl Pb challenging, as well as Cox2 protein level at 1.25 and 10 ug/dl Pb). Based on this comment, the sentence (201-202) “This effect was additionally dependent on the exposure time” is misleading.

Suggestions for the effect of time on COX-1 and COX-2 expression have been clarified.

178: “In our study 24-hour incubation did not significantly affect the protein expression of COX-1”. This is due to the low number of biological repetitions. For sure increasing the number of repetition will totally modify the statistical results, thus the meaning of the data. For this reason, the interpretation given in the paper is invalidated by an experimental bias.

According to Reviewer remark , it was indicated in the text that the results obtained may be the result of using too few repetitions.

Indeed (197-199) “Our results suggest that even very low Pb concentrations are able to enhance both transcription and translation of COX-1 in THP-1 macrophages” this sentence is true only looking at the 5 sand 10 Pb after 48h of challenging. If the induction of Cox1 at low Pb concentration would be the message (actually not demonstrated by the data), by adding some biological repetition and reaching the statistical relevance at 1.25, and 2.5 Pb at 24 h might be really important and interesting. Unlikely, the data as they are actually, do not confirms the sentences all long the text.

We deleted the sentence

180-181: “The increase in COX-2 protein expression was dose-dependent at each incubation time”. Not true. The increase is clearly dose-dependent at 24h, but the level of Cox2at 48h under 1.25 and 5 Pb are identical, with the intermediated dose of 2.5 showing a lower level. So, there is no “dose-dependence” at 48h.

According to Reviewer remark, it was noted that the increase in COX-2 protein expression proportional to the Pb dose used occurred only after 24 hours.

Despite the interest and the previous works of the Authors in neurodegenerative diseases, if the topic and, even more importantly all the data, are focused on blood Pb and macrophages, the vole paragraph from lane 205 to lane 212, has no sense here. At the best, this paragraph may be added to lines 185-197, as additional proof of the Cox1 and Cox2 role in inflammation. Where the sentence is located now interrupt the logical sequence of the discussion.

According to Reviewer remark we corrected paragraph.

The whole paragraph from line 213 to line224 looks excessive and not useful for the understanding and the discussion of the data obtained in the work. The following lines 255-230 are enough.

According to Reviewer remark we deleted and corrected paragraph.

234-235: “Our results indicate that even low concentrations of Pb cause an increase in production of inflammatory interleukins (IL-1β and IL-6) in THP-1 macrophages”. Based on the Figures of the paper, it is not true: only the higher concentration applied (in this work, 5 and 10 at 24h; 10 at 48h for both Il1beta and 6, plus 2.5 for Il6) are able to significantly inducing the cytokines. If the Authors refer to higher (in respect to the ones used here) concentrations used in published works, please refer to the relevant literature. Indeed, based on figure 4, Il6 never reach the statistical relevance vs the control.

According to Reviewer remark we corrected the sentence

237-239: “In addition, the longer incubation period intensified cytokine production, which was especially pronounced for IL-6 whose concentration after 48 hours of incubation with Pb was about 50 times higher than after 24 hours” despite true in terms of “crude numbers”, how do you explain that the Il6 expression in controls is also 50 and more time higher at 48h (about 600) vs 24h (bout 10)? If normalized vs the relative control (as indicated by the % at the bottom of each column) the level of IL6 at 24h is increased of about 60%, at 48h of about 50% (Pb 10).

According to Reviewer remark we corrected paragraph.

A246-253 “There is a growing belief that inflammation plays an important role in the development of CNS neurodegenerative diseases, i.e. Alzheimer's disease, amyotrophic lateral sclerosis, Parkinson's disease, and the prototypic neuroinflammatory disease, multiple sclerosis. Despite the different pathogenesis, the common feature of these diseases is chronic immune activation, especially of microglial cells [59,60,61]. Many studies have shown an increase in the expression of both cyclooxygenases and their products and proinflammatory interleukins in the course of neurodegenerative diseases [34,62,63,64]. In our study, we observed similar changes in response to very low Pb concentrations”. Again, if, as stated in the rebuttal, the aim of the study is the evaluation of the pro-inflammatory effect of low blood level concertation of Pf on circulating macrophages, this paragraph is out of theaim of the paper and more importantly far from the data here presented.

The reference to neuroinflammation has been removed from the discussion and conclusion.

Round 3

Reviewer 1 Report

Despite the Authors improved the description of the result’s section, the interpretation and discussion of the data is still weak. No correlation between the products of the activity of the Coxs enzymes and Cox1 vs Cox2 protein level is tempted. The same is true for the interleukins level and Cox1 vs Cox2 expression, thus the sentence “This clearly suggests that the development of inflammation is associated not only with COX-2 but also with COX-1” looks unsupported. A part for the style, no real improvements have been done in respect to the very first critiques.